# Understanding the infection severity and epidemiological characteristics of mpox in the UK

Thomas Ward [1] ✉, Christopher E. Overton[1,2], Robert S. Paton[1], Rachel Christie[1], Fergus Cumming[1] & Martyn Fyles[1]

In May 2022, individuals infected with the monkeypox virus were detected in the UK without clear travel links to endemic areas. Understanding the clinical characteristics and infection severity of mpox is necessary for effective public health policy. The study period of this paper, from the 1st June 2022 to 30th September 2022, included 3,375 individuals that tested positive for the monkeypox virus. The posterior mean times from infection to hospital admission and length of hospital stay were 14.89 days (95% Credible Intervals (CrI): 13.60, 16.32) and 7.07 days (95% CrI: 6.07, 8.23), respectively. We estimated the modelled Infection Hospitalisation Risk to be 4.13% (95% CrI: 3.04, 5.02), compared to the overall sample Case Hospitalisation Risk (CHR) of 5.10% (95% CrI: 4.38, 5.86). The overall sample CHR was estimated to be 17.86% (95% CrI: 6.06, 33.11) for females and 4.99% (95% CrI: 4.27, 5.75) for males. A notable difference was observed between the CHRs that were estimated for each sex, which may be indicative of increased infection severity in females or a considerably lower infection ascertainment rate. It was estimated that 74.65% (95% CrI: 55.78, 86.85) of infections with the monkeypox virus in the UK were captured over the outbreak.

The first human case of mpox was detected in a 9-month-old child from the Democratic Republic of the Congo (DRC) in 1970[1]. In early 2022, the virus was endemic in 12 countries within Africa[2] and had split into two distinct clades[3]. The true extent of the transmission of this virus was largely unknown, with divergent estimates of the case fatality risk suggesting considerable ascertainment bias in the sampling. In May 2022, cases of the monkeypox virus from non-endemic countries began to be detected without clear travel links to endemic areas. These cases were attributed to clade IIb, which is a subclade of the West African clade II. This outbreak has now spread to over 117 countries[4] and therefore understanding the clinical characteristics and infection severity is key to inform effective public health policy.

Early outbreaks of the monkeypox virus in the DRC were largely concentrated in children, with 52% of sampled cases analysed between 1980 to 1984[5] aged under 5. An analysis of 122 polymerase chain reaction (PCR) tests, during the 2017–2018 outbreak in Nigeria, found the average age had risen to 29 and 69% of cases were male[6]. The age composition of cases with the monkeypox virus will be influenced by public health policy, the circumstances of the outbreaks, changes in transmission routes, and vaccination campaigns. In the May 2022 pandemic, there had been a further increase in the median age of cases to 36 in the UK[7,8] and the proportion of cases that were male had risen to 99%. These cases were largely identified in dense interconnected networks, with the majority reporting multiple sexual contacts within the past 3 months, and 96.2% of cases identified as gay, bisexual, and other men who have sex with men (GBMSM)[9]. There is evidence to suggest that presymptomatic transmission may have facilitated the spread of the virus within these dense networks[10–13].

The case hospitalisation risk (CHR) and the case fatality risk (CFR) are the proportion of cases (infected individuals that have a positive

[1]UK Health Security Agency, Data Analytics & Surveillance, London, UK. [2]Department of Mathematical Sciences, University of Liverpool, Liverpool, UK.
✉ e-mail: Tom.Ward@ukhsa.gov.uk

diagnostic test), within a defined a temporal window, that are hospitalised or die (due to the infection), respectively. Substantial heterogeneity has been found in the CFR for monkeypox virus clades in Africa, ranging from 10.6% (95% Confidence Intervals (CI): 8.4, 13.3)[14] for clade I to 4.6% (95% CI: 2.1, 8.6)[14] for clade II. Prior to the global outbreak of clade IIb in 2022 the CFR for all mpox cases was estimated to be 8.7% (95% CI: 7.0, 10.8)[14] and until the 1990s all reported deaths that occurred were in children less than 10 years old. However, from 2000 the age distribution of deaths began to shift due to changes in the case composition and in the 2017-2018 outbreak in Nigeria the mean age of deaths recorded was 27 years old[6]. The risk of hospital admission prior to the 2022 epidemic was poorly characterised, which may have been influenced by public health policy and infrastructure limitations in the earlier outbreaks. From studies conducted in the DRC and the Central African Republic, CHR estimates have ranged from 73%[15] in 2003, 42% in 2005[16], and 83%[17] in 2015. The 2003 outbreak in the United States the CHR was estimated to be 26%[18] however, this varied between 10% to 69%[19] based on contact type.

Interpreting CHRs is challenging, because countries have distinct testing policies that can be temporally variable. Indeed, the severity bias, whereby the most severe cases will be those that seek healthcare or testing, can mean that CHRs are exaggerated, with milder cases omitted from the denominator. The infection hospitalisation risk (IHR) is the proportion of infected individuals that are hospitalised. To estimate the IHR, it is necessary to calculate the estimated incidence of infections based on the ascertainment rate (the proportion of infections that result

in a positive diagnostic test) with temporally corresponding hospitalisations, according to the time delay between infection and admission. In the UK, early estimates of the CHR were biased by a policy of clinical isolation of positive cases, independent of clinical need. However, this primarily affected the early stages of the outbreak, and the policy was removed when the outbreak became established.

To understand the burden of infection from the monkeypox virus we have calculated key epidemiological metrics within the UK. The study used a Bayesian doubly interval censored model adjusted for right truncation to calculate the time from infection to hospital admission, infection to a first positive test, and the length of hospital stay. The instantaneous and overall CHR are calculated, further subset by sex and age. The IHR is calculated through a Bayesian modelling approach that estimated the ascertainment rate of infections through the use of contact tracing data over the outbreak.

## Results
### Infection to hospitalisation
The posterior estimate for the mean time from infection to hospital admission was 14.89 days (95% Credible Intervals (CrI): 13.60, 16.32) (Fig. 1 and Table 1). The lognormal distribution had the lowest Leave One Out (LOO) cross validation score (Supplementary Table 1) although there was not strong evidence of a difference in the out of sample error relative to the other distributions analysed. The full results for each distribution can be seen in Supplementary Table 2. The estimated values of the cumulative distribution function for the time

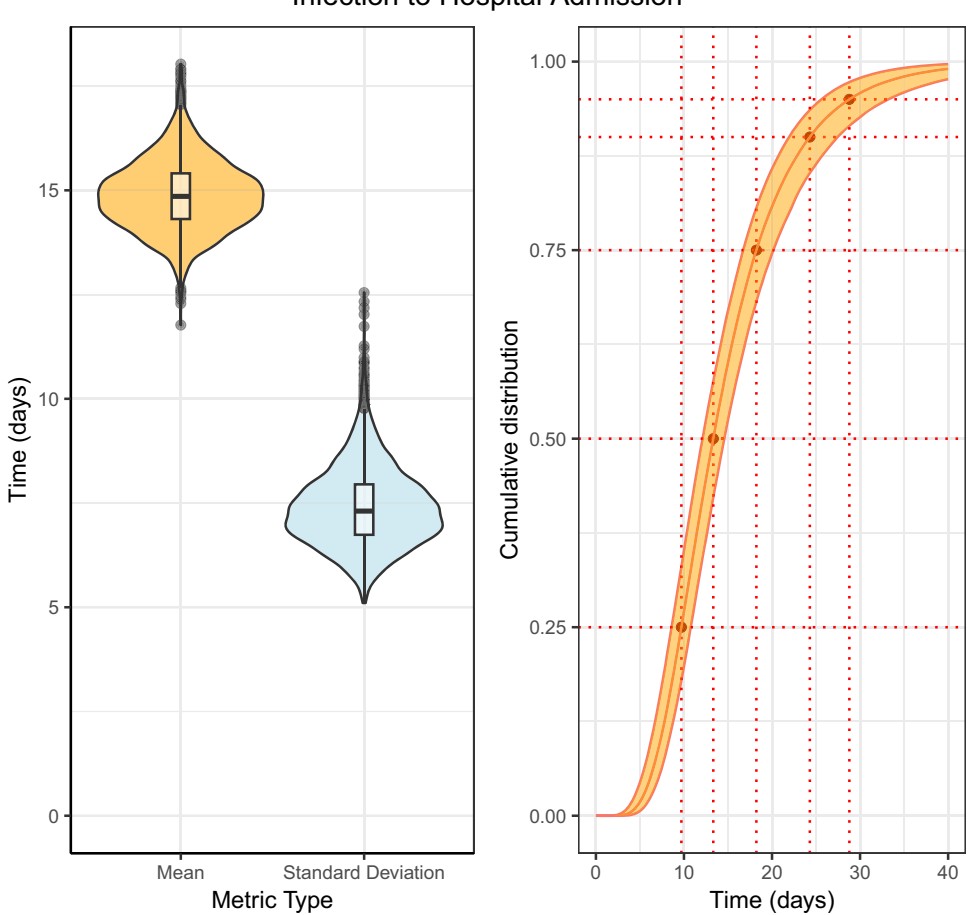

## Infection to Hospital Admission

**Fig. 1 | Posterior distribution for the time from infection to hospital admission fit to the data of 118 cases using a lognormal distribution.** Left: a violin plot of the mean and standard deviation, with a box and whisker plot including the minima, 1st quartile, median, 3rd quartile, and maxima of the posterior distribution. Right: the cumulative distribution function with 95% credible intervals.

from infection to hospital admission can be seen in Supplementary Table 3, the median was 13.33 days (95% CrI: 12.11, 14.58) and at the 95th percentile the posterior estimate was 28.79 days (95% CrI: 25.62, 33.21). The hospital admission dates of patients with mpox that had a recorded symptom onset and exposure date (and histograms of the distributions) can be seen in Supplementary Figs. 1, 2.

### Infection to first positive test

The posterior estimate for the mean time from infection to first positive test was 15.14 days (95% CrI: 13.75, 16.65) (Fig. 2 and Table 2). The gamma distribution had the lowest LOO cross validation score however, the results are not strong evidence that that gamma had substantially lower average out-of-sample error from the Weibull or lognormal distributions. The full results for each distribution can be

seen in Supplementary Table 4. The estimated values of the cumulative distribution function for the time from infection to first positive test can be seen in Supplementary Table 5, the median was 13.77 days (95% CrI: 12.36, 15.19) and at the 95th percentile the posterior estimate was 29.70 days (95% CrI: 26.80, 33.62). The data for the time from infection to first positive test can be seen in Supplementary Fig. 3.

### Length of stay

The posterior estimate for the mean length of stay in hospital was 7.07 days (95% CrI: 6.07, 8.23) (Fig. 3 and Table 3). The lognormal distribution had the lowest LOO cross validation score however, the results (Supplementary Table 1) are not strong evidence that the lognormal model had substantially lower average out-of-sample error than the gamma or Weibull distributions. The full results for each distribution can be seen in Supplementary Table 6. The estimated values of the cumulative distribution function for the length of stay can be seen Supplementary Table 7, the median was 4.03 days (95% CrI: 3.52, 4.60) and at the 95th percentile the posterior estimate was 22.84 days (95% CrI: 18.89, 28.12). The hospital admission dates of patients with mpox and a histogram of the individual lengths of stay can be seen in Supplementary Fig. 4.

### Case hospitalisation risk

There was limited variation in the instantaneous CHR across the study period with point estimates that largely overlap (Fig. 4). We identified distinct sex-specific sample CHRs (Table 4) with females found to have

**Table 1 | Summary statistics of the time from infection to hospital admissions, fit to data from 118 cases using a log-normal distribution**

| Infection to hospital admission | | | | | | |
|---|---|---|---|---|---|---|
| N | Distribution | Mean | Standard Deviation | Shape/ location | Scale | $\hat{R}$ (Mean) |
| 118 | Interval censoring right truncation corrected - lognormal | 14.89 (13.60, 16.32) | 7.40 (6.01, 9.11) | 2.59 (2.49, 2.68) | 0.47 (0.40, 0.54) | 1.00 |

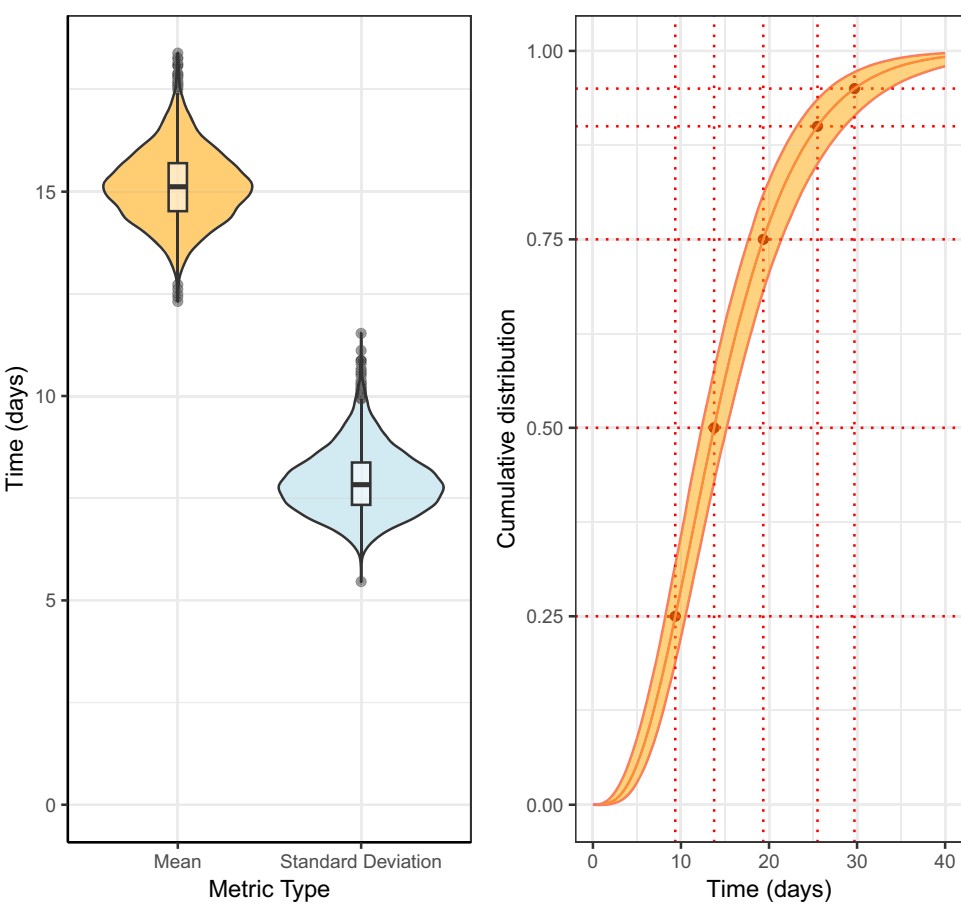

### Infection to First Positive Test

**Fig. 2 | Posterior distribution for the time from infection to first positive test fit to the data of 86 cases using a gamma distribution.** Left: a violin plot of the mean and standard deviation, with a box and whisker plot including the minima, 1st quartile, median, 3rd quartile, and maxima of the posterior distribution. Right: the cumulative distribution function with 95% credible intervals.

a far higher risk of hospital admission. However, there is considerable uncertainty due to the small number of female cases in the clinical and surveillance data. The age distribution of the admissions and the sample CHR by age groups can be seen in Supplementary Figs 5 and Supplementary Fig. 6, respectively.

## Infection hospitalisation risk

Understanding the ascertainment rate of infections for an infectious disease can be difficult because infections with less severe or sub-clinical levels of disease may not be identified[20]. Whereas individuals that seek medical attention or diagnostic testing may be on average more severe. Therefore, to estimate the ascertainment rate, we first examined the difference between the CHRs of case subgroups, which were conditional upon notification of exposure (please see the Methods section).

An index infection is defined, for this paper, as an infected individual that did not receive a notification of their exposure to the monkeypox virus, and an index case is an individual that did not receive a notification of their exposure and was identified through their presentation to a clinician or healthcare provider. Conversely, a secondary infection is defined as an infected individual that was notified of their exposure, and if a secondary infection was subsequently tested for the monkeypox virus then the individual became a secondary case.

The modelled CHR varied between 8.47% (95% CrI: 7.08, 9.97) and 4.55% (95% CrI: 3.81, 5.25) for the index and secondary cases, respectively. We observe that the CHR of secondary cases is approximately half that of index cases, leading to the constraint that the ascertainment of notified cases is approximately twice as high as the ascertainment of cases that were not contact traced or notified of exposure. For untraced cases, it was unknown whether the individual was aware of their exposure status, and these cases did not appear in our contact tracing data, and consequently could not be assigned either index or secondary case status. However, as a result of estimating the composition of the untraced group in terms of whether they were aware of their exposure or not i.e., index or secondary case status, we obtained larger effective sample sizes for each of the exposure groups. This allowed us to produce improved estimates for the CHR of each group, which we provide in Supplementary Table 8.

We use the calculated differences between the CHRs of the index/secondary/untraced cases to constrain the range of plausible ascertainment rates and therefore plausible estimates of the IHR, which we

**Table 2 | Summary statistics of the time from infection to first positive test, fit data from 86 cases using a gamma distribution**

| Infection to first positive test | | | | | | |
|---|---|---|---|---|---|---|
| N | Distribution | Mean | Standard Deviation | Shape/ location | Scale | $\hat{R}$ (Mean) |
| 86 | Interval censoring right truncation corrected - gamma | 15.14 (13.75, 16.65) | 7.90 (6.75, 9.31) | 3.74 (2.85, 4.73) | 0.25 (0.18, 0.32) | 1.00 |

## Length of Stay

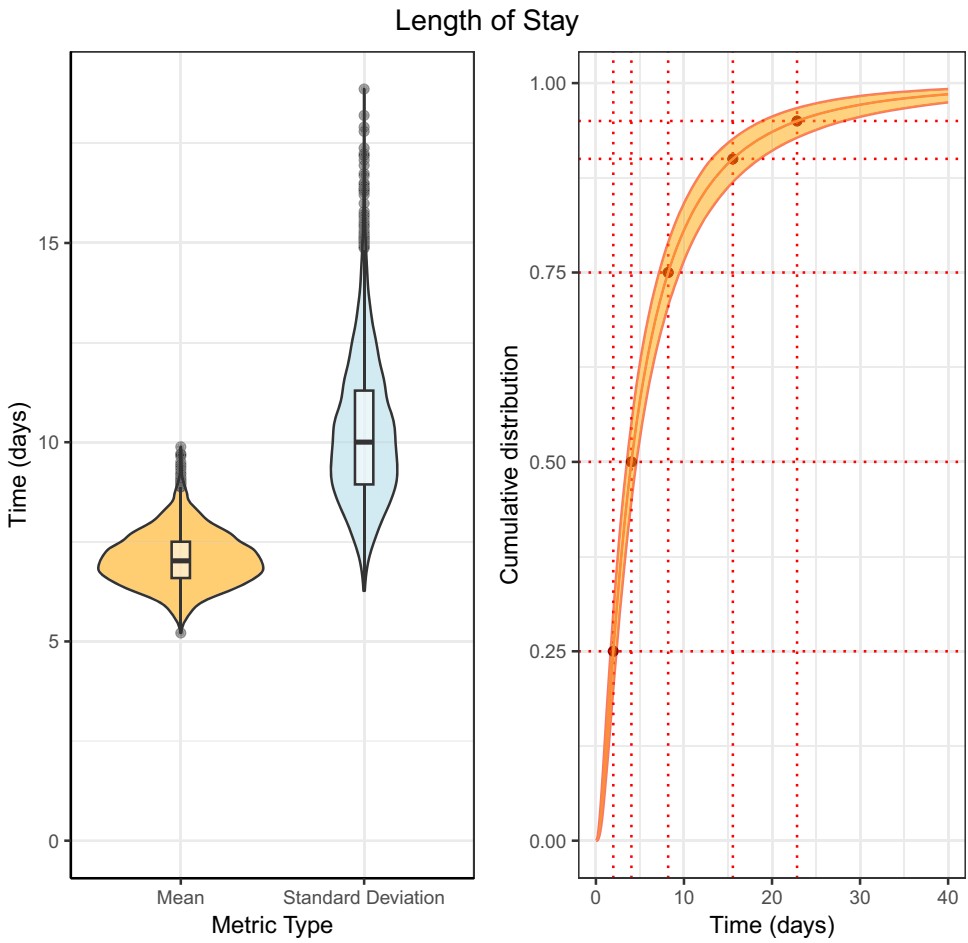

**Fig. 3 | Posterior distribution for the hospital length of stay fit to the data of 155 patients using a lognormal distribution.** Left: a violin plot of the mean and standard deviation, with a box and whisker plot including the minima, 1st quartile, median, 3rd quartile, and maxima of the posterior distribution. Right: the cumulative distribution function with 95% credible intervals.

refer to as the case ascertainment bias prior. The obtained infection ascertainment bias prior distribution (Supplementary Fig. 7) has a linear relationship between the ascertainment rate of index and secondary infections, which is necessary to model the observed bias in the CHR between index and secondary cases.

In Supplementary Table 9 we report the posterior estimates for the probability of ascertaining an index, secondary, and untraced infection. Our model estimated that roughly half of all individuals who were not made aware of their exposure via contact tracing were ascertained 49.13% (95% CrI: 34.80, 63.95). While the majority who were aware of their exposure due to having been contact traced were captured 90.84% (95% CrI: 70.18, 99.74). Additionally, we estimated, based on the severity profile, that infections which were not made aware of their exposure were only a small proportion of the untraced population 10.08% (95% CrI: 0.31, 32.80).

After modelling the ascertainment rate, we then adjust the observed case time series for the estimated ascertainment rates to produce the infection incidence time series. We use the estimated time delay distributions for the time to detection and hospitalisation to temporally fit the estimated incidence and observed hospitalisations. The model calculates the expected number of admissions for each exposure group based on the estimated temporal incidence. The

temporal probabilities for the ascertainment of the infections in each exposure subgroups are then adjusted iteratively based on how well the expected hospitalisations match the observed hospitalisations. If the estimated incidence aligns with the observed hospitalisations, it indicates that the model is capturing the dynamics of incidence accurately. The adjustment for the ascertainment probabilities aims to find a balance where the model-predicted number of infections corresponds closely to the hospitalisations. Therefore, the model is employing a feedback loop, adjusting ascertainment probabilities to achieve a coherent relationship between estimated incidence, expected hospitalisations, and observed hospitalisations. In essence, iteratively optimising to enhance the precision of its predictions in response to real-world hospitalisation outcomes. This iterative process refines the model's understanding of how infections are ascertained and recorded in the context of hospitalisations rather than simply using estimated cumulative totals. This allows us to gain further insight into the ascertainment rates and consequently the IHR.

We estimated the modelled IHR to be 4.13% (95% CrI: 3.04, 5.02), compared to the overall sample CHR with binomial uncertainty of 5.10% (95% CrI: 4.38, 5.86). The posterior distribution of the IHR, the overall ascertainment rate, the estimated total number of infections, and the estimated total number of non-ascertained infections can be seen in Fig. 5. The modelling estimates that 74.65% (95% CrI: 55.78, 86.85) of infections over the period analysed in the UK were ascertained (Table 5).

The modelled incidence rates over time for each subgroup (Supplementary Fig. 8) show that there was approximately zero incidence for both index and secondary cases after the July midpoint, due to changes in contact tracing. After this point, all cases were defined as untraced, as we did not know whether these cases were notified of exposure. For the untraced cases, we observe a dramatic reduction in the incidence rate towards the end of August.

The posterior has significantly reduced uncertainty compared to the prior (Supplementary Figs. 9, 10). In particular, the model excludes the possibility of a low ascertainment rate of secondary cases and the

**Table 3 | Summary statistics of the length of stay in hospital, fit to data from 155 patients using a lognormal distribution**

| Length of hospital stay | | | | | | |
|---|---|---|---|---|---|---|
| N | Distribution | Mean | Standard Deviation | Shape/ location | Scale | $\hat{R}$ (Mean) |
| 155 | Interval censoring right truncation corrected - lognormal | 7.07 (6.07, 8.23) | 10.22 (7.73, 13.36) | 1.39 (1.25, 1.53) | 1.06 (0.96, 1.16) | 1.00 |

## Case Hospitalisation Risk (CHR) − Instantaneous and Overall

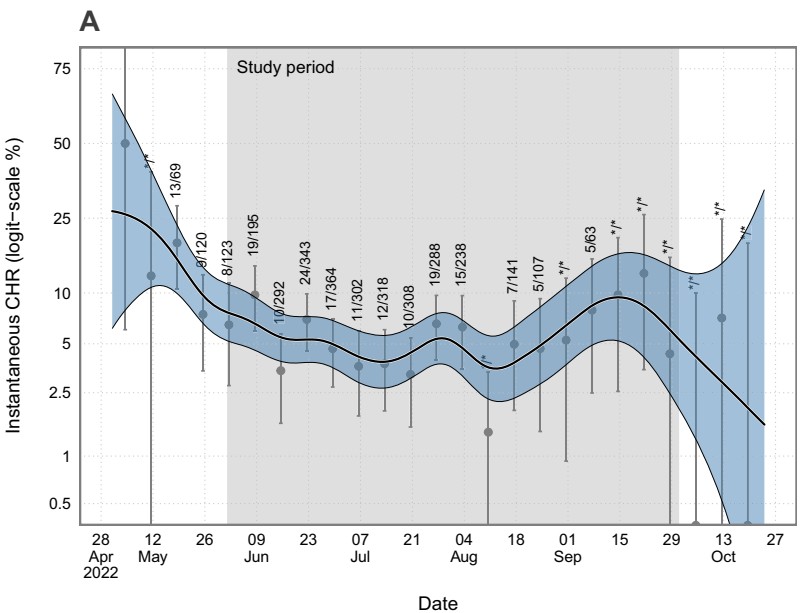
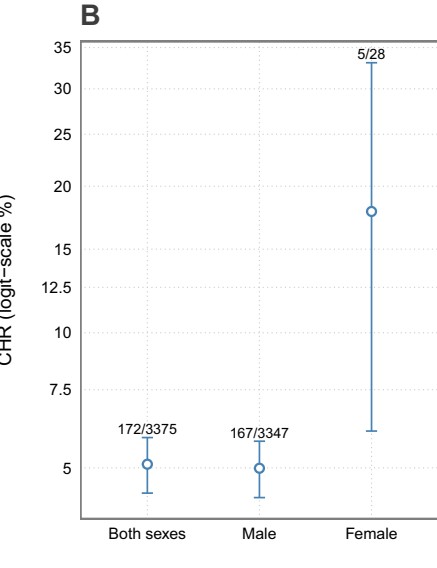

**Fig. 4 | The instantaneous and overall Case Hospitalisation Risk (CHR). A** The estimated instantaneous sample CHR. The grey points represent weekly CHRs, with 95% binomial credible intervals (some values of N are not displayed to comply with data privacy guidance). The black line and ribbon show the mean and 95% confidence intervals of a logistic generalised additive model on the probability of hospital admission. A shaded grey region shows the range of specimen dates considered in our analysis (1 June 2022 to 30 September 2022). **B** The overall and sex-specific sample CHRs with 95% binomial credible intervals.

possibility of an IHR lower than approximately 2.5%. In the prior distribution, there is a strong linear correlation between the probability of hospitalisation and the ascertainment rate of index and secondary infections, however the posterior density has concentrated around a singular point. Therefore, we conclude that through estimation of the incidence rate, and fitting to time series data, we have resolved the non-identifiability of the IHR and the ascertainment of cases. The credible intervals of the posterior predictive distribution contain nearly all the observed datapoints, as is desired. This suggests that the model performs well at describing the data generation process.

In Supplementary Figs. 11, 12 we model the IHR when the untraced group is not included, which we performed as further sensitivity analysis. In comparison to the infection ascertainment bias prior with the untraced cases included, there is significantly more uncertainty in the ascertainment rate of index and secondary infections (Supplementary Table 10). This is because when the untraced cases are included, there is a larger effective sample size. Both the infection ascertainment priors obtained when including or excluding untraced cases share a non-identifiability in the IHR and the ascertainment rate of non-hospitalised infections. The difference between the prior and posterior is relatively small, however it tends towards the posterior we obtained when we included the untraced cases. The inclusion of untraced cases, therefore, provides a greatly increased effective sample size, and without using the untraced cases, there is significantly increased uncertainty in the obtained posteriors. In Supplementary Figs. 13–16 we provide detailed comparisons between the prior and posterior densities for the key parameters of interest when the untraced sub-population is included or excluded.

## Discussion

There have been 90,439 laboratory confirmed cases[21] of mpox since the epidemic began in May 2022[22]. We estimated that, in the UK, the instantaneous CHR varied between 3.53% (95% CI: 2.20, 5.60) to 9.43% (95% CI: 5.18–16.57) as public health testing infrastructure, messaging and policy evolved. Considerable variation was found in the estimated CHR for each sex, which may be indictive of increased severity or a substantial reduction in the ascertainment rate of female infections. We estimate the IHR to be 4.13% (95% CrI: 3.04, 5.02), with an infection ascertainment rate of 74.65% (95% CrI: 55.78, 86.85).

The CHR can be calculated through various methodological approaches, which can confound comparisons between different studies. Two general approaches to calculate the CHR can be defined as the overall CHR and the instantaneous CHR[23]. The overall CHR is the proportion of cases that were hospitalised up to a certain date or within a specific study period. The overall CHR can therefore be considered as the average hospital burden over an epidemic. The instantaneous CHR is the real time proportion of cases that are hospitalised over time, usually measured within days or weeks. These approaches for calculating the CHR can be estimated from aggregate or row level/linelist data on individuals. Individual level data requires unique identifiers for data linkage between cases and hospital admissions, which in many studies is not available. Aggregate data approaches can only approximate the CHR and require a time varying delay period adjustment, as the case composition changes, for the time from test report date to hospital admission. These methods are impacted by right censoring at the end or beginning of a study period, because confirmed cases may not yet have been hospitalised, and can be adjusted for using threshold criteria that incorporates the time from the detection of a case to a hospital admission. This bias was observed

**Table 4 | The overall and sex-specific sample CHR across the study period with 95% binomial credible intervals**

|  | Hospitalised | Total | Sample CHR (%, 95% CrIs) |
|---|---|---|---|
| Total | 172 | 3375 | 5.10 (95% CrI: 4.38, 5.86) |
| Male | 167 | 3347 | 4.99 (95% CrI: 4.27, 5.75) |
| Female | 5 | 28 | 17.86 (95% CrI: 6.06, 33.11) |

## Posterior Densities

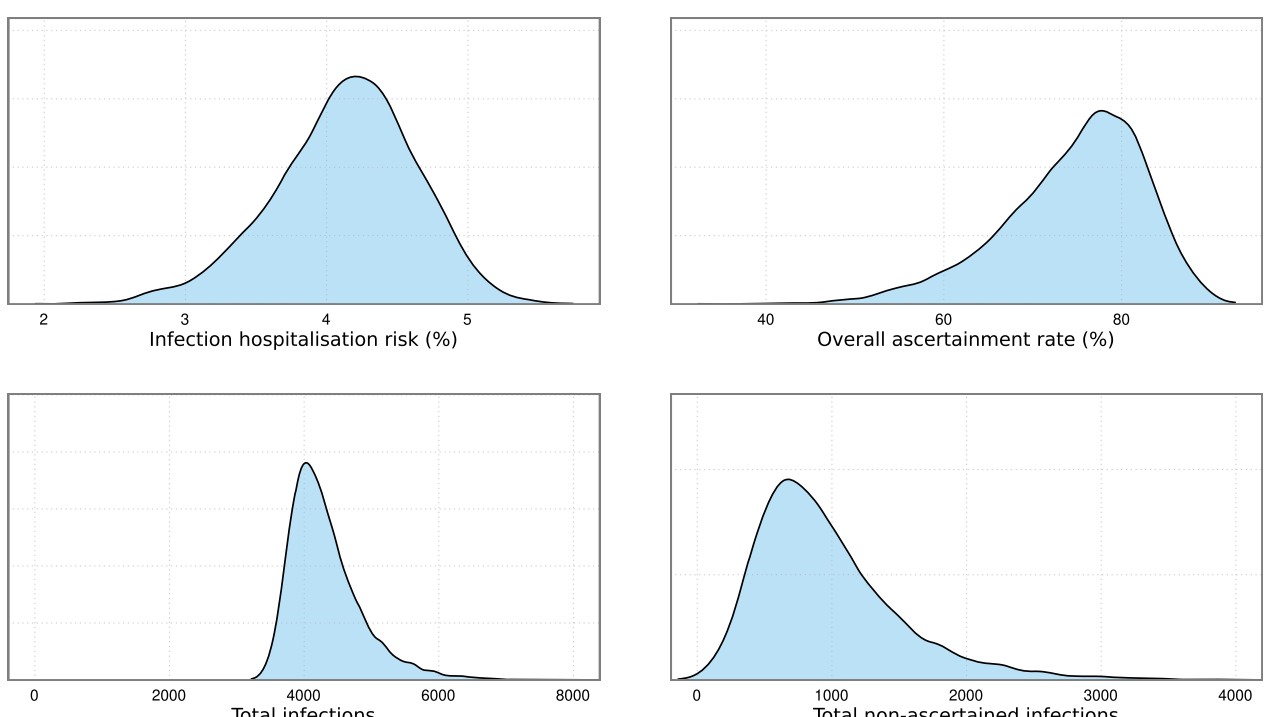

**Fig. 5 | The posterior density for several key parameters of interest:** the infection hospitalisation risk, the overall ascertainment rate, the total number of infections and the total non-ascertained infections.

**Table 5 | Table of the posterior density for several key parameters of interest: the overall ascertainment rate, the total number of infections and the total non-ascertained infections**

| Quantity | Estimate |
|---|---|
| Infection Hospitalisation Risk (%) | 4.13 (95% CrI: 3.04, 5.02) |
| Overall ascertainment rate (%) | 74.65 (95% CrI: 55.78, 86.85) |
| Total infections | 4329 (95% CrI: 3596, 5714) |
| Total non-ascertained infections | 976 (95% CrI: 243, 2361) |

for early estimates of COVID-19 in Wuhan where the cumulative number of outcomes were divided by total cases however, when adequate adjustment was included for censoring, the estimates were considerably higher. Due to spatial and temporal changes in testing rates, public health messaging, and population immunity there are limitations to the interpretation of an overall CHR. The instantaneous CHR is therefore a better reflection of the real time risk whereas the overall CHR captures the impact on the population of interest.

There has been considerable inter-country heterogeneity in the estimated CHRs across the 2022 outbreak of the monkeypox virus, which will have been influenced by differences in policy, such as clinical isolation, and public health messaging. These distinctions in policy will have an impact on the case ascertainment rate and the criteria for hospital admission[24]. Studies published on this outbreak have presented overall CHR estimates[25–32] and have not stated the inclusion of an adjustment for right censoring at the end of the study period. Therefore, these methods can be described as crude approximations to the overall CHR. The CHR has ranged from 2%[25], from a sample of 181 cases in Spain, to 13%[26] from a study of 528 cases across 16 countries from April to June. The largest report from the ECDC and WHO[27] found the crude CHR to be 6.7%, which included data from 41 countries.

Earlier outbreaks of the monkeypox virus in the DRC and West Africa were limited by the public health infrastructure. Cases that were identified in non-endemic countries, prior to 2022, were also in many instances precautionarily hospitalised for observation and isolation. This continued at the outset of the 2022 outbreak and was a policy that has been subsequently abandoned in the UK. Therefore, defining a hospital admission due to the severity of an infection becomes complicated by these biases and, as a result, past and current estimates of the CHR are not comparable measures of infection severity. The ascertainment of cases can be further understood through analysis of the success rate of contact traced individuals and the difference in the severity profiles of index relative to secondary cases[20].

In this study we have calculated an overall IHR across the outbreak. However, the varying admission criteria for mpox at each hospital could have affected the IHR. In the initial stages of the outbreak, all cases were admitted to hospital for isolation. As the outbreak grew, this policy was lifted. However, a small proportion of individuals, prior to its derogation as a high consequence infectious disease[33], were still admitted for isolation purposes. To reduce the influence of changing isolation policies on our analysis, we limited our study period to the time when the instantaneous CHR was roughly constant. The estimated IHR will have been impacted by the likelihood of an individual seeking Secondary Care treatment, which is influenced by public health messaging and the perceived risk of an infection. The average length of stay for an mpox patient may also be affected by public health guidance on containment and observation as well as patient severity. To limit this bias, we removed individuals that presented at hospital for diagnostic testing.

The study was limited by the short period of time that contact tracing was conducted. The difference between severity of index and secondary cases, as defined in this study, is the main source of information on severity and ascertainment. However, this can only be calculated from the contact tracing data, which

was not widely collected after the 1st August 2022. A relative change in severity following this period would therefore be more difficult to detect. However, as the CHR was roughly constant, this is unlikely to be the case. In this study, parametric distributions were assumed for the time delay distributions. We used model evaluation scores to assess the distributions considered. The impact of age and sex on the CHR were also investigated. However, other factors such as HIV status, would be important determinants for hospitalisation risk to investigate further.

The global spread of the monkeypox virus has caused international public health concern and considerable healthcare burden for the affected countries. We estimated the mean time from infection to hospital admission to be 14.89 days (95% CrI: 13.60, 16.32) and the average length of hospital stay was 7.07 days (95% CrI: 6.07, 8.23). We found considerable variation for the sex-specific CHR. The observed increased risk of hospital admission for female cases may be indicative of the diminished ascertainment of infections or increased severity. The overall IHR was calculated to be 4.13% (95% CrI: 3.04, 5.02) and we estimated that around 75% infections from the monkeypox virus in the UK were captured over the study period. Ongoing public health surveillance should therefore be conducted to limit the healthcare burden on the affected countries.

## Methods
### Epidemiological data
Cases of the monkeypox virus were monitored by the UKHSA using testing data from affiliated laboratories and NHS laboratories, contact tracing, and case questionnaires (collected by UKHSA health protection teams). A confirmed case is an individual with a positive PCR test result for the monkeypox virus, and a highly probable case is an individual with a positive PCR test result for orthopoxvirus. As of 25th July 2022, both definitions were recognised in the UK to represent a case of mpox.

Hospital episode statistics for inpatients were obtained from the NHS Digital Secondary Uses Services data set[34] and A&E attendances were obtained from the Emergency Care Data Set[35], both datasets contain clinical, patient, administrative and geographic information about patient admissions. A&E attendances and inpatient records were extracted, and hospital episodes linked by an NHS identifier to a positive test result. An admission for mpox is defined as a patient having one of the following:

- An mpox diagnosis code (B04)[36].
- A positive PCR test for the monkeypox virus within 21 days after admission to hospital.
- Tested positive for the monkeypox virus during a hospital admission.

### Study period
We restricted the study period to all cases that tested positive for the monkeypox virus and had qualifying hospital episodes with mpox from 1st June 2022 to 30th September 2022. This study period is applied to the time delay estimations, IHR, and CHR, to ensure estimates are temporally consistent.

### Data preparation
Data was extracted on the 26th of October 2022, at which time 3776 people had tested positive for the monkeypox virus in the UK, 3375 of which had specimen date within the study period, and 172 of those had an associated hospital episode of 1 day or longer. Length of stay was calculated from the date of admission to the date of discharge and patients were excluded if the discharge date was missing. This is due to the nature of the hospital admissions data, where records are only reported after discharge, so the discharge dates for these patients will be incorrectly missing rather than representing patients still in hospital. This resulted in 155 patients being suitable for the length of stay

analysis. Supplementary Fig. 17 shows a flowchart with the number of cases excluded at each stage.

To calculate the time from infection to hospitalisation, patients that reported a symptom onset date after hospitalisation were excluded (since this likely corresponded to an incorrect symptom onset date in the questionnaire data) and for those that had multiple admissions the earliest admission date was used. The symptom onset date was identified through contact tracing conducted by UKHSA health protection teams and questionnaires completed by the cases (via the question 'On what date did your illness begin?'). This definition of symptom onset describes the date that an individual first noticed their symptoms; though the true date of symptom onset could have been earlier but not detected. Of the 3776 cases that tested positive for the monkeypox virus, 2360 had symptom onset information, of which 110 had a hospital admission. Analysis was also conducted to measure the time from infection to hospitalisation for patients that had exposure dates reported. Exposure date was identified through cases which had completed a questionnaire and answered the questions 'In the 21 days (3 weeks) before first symptom onset did you have contact with anyone with suspected or confirmed monkeypox infection?' and 'Date of last contact with case'. 92 patients of the 3776 reported an exposure date, 8 of which had a hospital admission. Then of the 92 patients with exposure date, 86 had a reliable specimen collection date, and were used for estimating the delay from infection date to first positive test.

In Supplementary Table 11 we report for each group the sample size, the mean age, the proportion by sex, and proportion of cases reported as GBMSM. Supplementary Figs. 1–4 show histograms of the time delay data for symptom onset to hospital admission, exposure to hospital admission, exposure to first positive test, and length of stay, respectively.

## Time delay modelling

**Infection to hospital admission.** To model the risk of an infection becoming hospitalised, we first need to measure the time delay between infection and hospitalisation. However, infection events are rarely observed directly. Symptom onset date, conversely, was reported for 2360 out of 3776 cases confirmed through PCR testing. By combining this data with estimates of the incubation period from the literature[10], we can estimate the time delay distribution from infection to hospital admission.

The data on symptom onset date and hospital admission date only provide the date of the event, rather than the time. Therefore, the time of each event is interval censored – we know the date on which the event occurred but not the precise time. As a result, each data point consists of a pair of observation intervals, $[o_1, o_2]$ for symptom onset time, represented by a real valued random variable $O$, and $[h_1, h_2]$ for hospital admission time, represented by a real valued random variable $H$. Since the data are daily censored, we have intervals $[o_1, o_1 + 1]$ and $[h_1, h_1 + 1]$, where $o_1$ and $h_1$ are integers representing the observed onset date and admission date, respectively.

In our full data, we observe symptom onset dates once an individual tests positive. However, we do not know which individuals will be admitted to hospital *apriori*, so we cannot treat the data as right-censored, and instead an individual only enters our data set after they are admitted to hospital, which leads to right truncation in the data, where $T$ is an integer denoting the truncation date. Right-truncation leads to the observed time delays being shorter than the true time delays[37], since recently infected individuals will only be in the data if they had a relatively short time delay from infection to hospital admission.

To look at the time since infection, rather than time of symptom onset, we need to account for the incubation period of each individual, which we denote by a real valued random variable $D$. However, the incubation period is not observed directly, so we have interval censoring between 0, the minimum incubation period, and infinity, the

maximum incubation period. From Ward et al.[10], we know the incubation period distribution, which we can use to inform the likelihood function. Our full likelihood function, accounting for the right truncation and interval censoring, is given by

$$P(h_1 < H < h_2 | o_1 < O < o_2, H < T, 0 < D < \infty) = \frac{P(h_1 < H < h_2, o_1 < O < o_2, 0 < D < \infty)}{P(H < T, o_1 < O < o_2, 0 < D < \infty)}$$
$$= \frac{\int_{o_1}^{o_2} \int_{h_1}^{h_2} \int_0^{\infty} f(H = h, O = o, D = d) \, dd \, dh \, do}{\int_{o_1}^{o_2} \int_o^T \int_0^{\infty} f(H = h, O = o, D = d) \, dd \, dh \, do}$$
(1)

Evaluating this full likelihood is computationally expensive. Therefore, we consider an approximate latent variable approach, following ref. 10. We introduce three random variables, $o^*$, the time of symptom onset within the observation window, $h^*$, the time of hospital admission within the observation window, and $d^*$, the length of the incubation period. For the two event times ($o^*$ and $h^*$), we assume uniform prior distributions over the observation window, and for the incubation period, $d^*$, we assume a prior distribution taken from the incubation period distribution estimated in ref. 10. That is,

$$o^* \sim \text{Uniform}(o_1, o_2),$$
$$h^* \sim \text{Uniform}(h_1, h_2),$$
$$d^* \sim \text{Weibull}(1.4, 8.5).$$

Using these latent variables our likelihood function becomes

$$P\left(H = h^* | O = o^*, H < T, D = d^*\right) = \frac{P\left(H = h^*, O = o^*, D = d^*\right)}{P\left(O = o^*, H < T, D = d^*\right)} = \frac{P\left(H = h^*, I = o^* - d^*\right)}{P\left(H < T, I = o^* - d^*\right)}$$
$$= \frac{P\left(H = h^* | I = o^* - d^*\right)}{P\left(H < T | I = o^* - d^*\right)} = \frac{f_\theta\left(h^* + d^* - o^*\right)}{F_\theta\left(T + d^* - o^*\right)},$$
(2)

where $I$ is the time of infection and $f_\theta$ and $F_\theta$ are the probability density and cumulative density functions of the infection to hospitalisation delay, respectively, parameterised by parameters $\theta$. This latent variable approach is an approximation since it assumes independence between the incubation period and symptom onset date. In reality, the incubation period is subject to an epidemic phase bias[38], whereby the observed incubation periods depend on the symptom onset date. However, the incubation period is shorter than the delay from symptom onset date to hospital admission, so this bias should only have a small effect on the estimated time from infection to hospital admission. To accurately correct for this bias, one would need to sample the latent incubation period variable from a prior distribution conditional on symptom onset.

In addition to date of symptom onset, for a subset of patients we have the date of exposure. Therefore, for these individuals the data takes the form of a pair of observations: an infection date $I \in [i_1, i_2]$ and admission date $H \in [h_1, h_2]$. Since infection date is observed, we do not need to consider the incubation period for these data points, so we have likelihood function

$$P(h_1 < H < h_2 | i_1 < I < i_2, H < T) = \frac{P(i_1 < I < i_2, h_1 < H < h_2)}{P(H < T, i_1 < I < i_2)}$$
$$= \frac{\int_{i_1}^{i_2} \int_{h_1}^{h_2} f(H = h, I = i) \, dh \, di}{\int_{i_1}^{i_2} \int_i^T f(H = h, I = i) \, dh \, di}.$$
(3)

To handle the interval censoring, we again consider a latent variable approach,

$$i^* \sim \text{Uniform}(i_1, i_2),$$
$$h^* \sim \text{Uniform}(h_1, h_2),$$
$$P\left(H = h^* | I = i^*, H > T\right) = \frac{P(I = i^*, H = h^*)}{P(H < T, I = i^*)} = \frac{f_\theta(h^* - i^*)}{F_\theta(T - i^*)}. \quad (4)$$

Therefore, our full likelihood function for modelling the infection to hospital admission delay is given by

$$o^* \sim \text{Uniform}(o_1, o_2),$$
$$h^* \sim \text{Uniform}(h_1, h_2),$$
$$d^* \sim \text{Weibull}(1.4, 8.5),$$
$$i^* \sim \text{Uniform}(i_1, i_2),$$
$$P\left(H = h^* | I = i^*, H > T\right) = \frac{f_\theta(h^* - i^*)}{F_\theta(T - i^*)}, \quad (5)$$
$$P\left(H = h^* | O = o^*, H < T, D = d^*\right) = \frac{f_\theta(h^* + d^* - o^*)}{F_\theta(T + d^* - o^*)}.$$

**Infection to positive test.** In the IHR model that we develop, one of the parameters is the time delay distribution from individuals becoming infected and returning their first positive test. We will call this the ascertainment delay. For the 92 cases with a recorded exposure date, 86 of these have a known positive test date after their exposure date. We have interval censored data for exposure time, $E \in [e_1, e_2]$, and positive test time, $\tau \in [\tau_1, \tau_2]$, which are right truncated by time $T$. We have the likelihood function

$$P\left(\tau_1 < \tau < \tau_2 | e_1 < E < e_2, \tau < T\right) = \frac{P(\tau_1 < \tau < \tau_2, e_1 < E < e_2)}{P(\tau < T, e_1 < E < e_2)}$$
$$= \frac{\int_{e_1}^{e_2} \int_{\tau_1}^{\tau_2} f(E = e, \tau = \tau') d\tau' de}{\int_{e_1}^{e_2} \int_{e}^{T} f(E = e, \tau = \tau') d\tau' de}. \quad (6)$$

Similarly, to the infection to hospitalisation delay, we consider a latent variable approach,

$$e^* \sim \text{Uniform}(e_1, e_2),$$
$$\tau^* \sim \text{Uniform}(\tau_1, \tau_2),$$
$$P\left(\tau = \tau^* | E = e^*, \tau > T\right) = \frac{P(\tau = \tau^*, E = e^*)}{P(\tau < T, E = e^*)} = \frac{w_\theta(\tau^* - e^*)}{W_\theta(T - e^*)}, \quad (7)$$

where $w_\lambda$ and $W_\lambda$ the probability density and cumulative density functions of the ascertainment delay distribution, respectively, parameterised by parameters $\theta$.

**Hospital length of stay.** To understand the impact on the healthcare system, we also need to model the delay from hospital admission to leaving hospital (also referred to as length of stay). As above, we have interval censored data for hospital admission time, $H \in [h_1, h_2]$, and for hospital leaving time, $L \in [l_1, l_2]$. These data are also subject to right truncation, $L < T$, but we do not need to consider any other delays in this model, since we are interested in the time delay between $H$ and $L$. Therefore, we have the likelihood function

$$P\left(l_1 < L < l_2 | h_1 < H < h_2, L < T\right) = \frac{P(l_1 < L < l_2, h_1 < H < h_2)}{P(L < T, h_1 < H < h_2)}$$
$$= \frac{\int_{h_1}^{h_2} \int_{l_1}^{l_2} f(H = h, L = l) dl dh}{\int_{h_1}^{h_2} \int_{h}^{T} f(H = h, L = l) dl dh}. \quad (8)$$

Similarly, to the infection to hospitalisation delay, we consider a latent variable approach,

$$l^* \sim \text{Uniform}(l_1, l_2),$$
$$h^* \sim \text{Uniform}(h_1, h_2),$$
$$P\left(L = l^* | H = h^*, L > T\right) = \frac{P(L = l^*, H = h^*)}{P(L < T, H = h^*)} = \frac{g_\theta(l^* - h^*)}{G_\theta(T - h^*)}, \quad (9)$$

where $g_\lambda$ and $G_\lambda$ the probability density and cumulative density functions of the length of stay distribution, respectively, parameterised by parameters $\theta$.

**Prior distributions.** For all three time delay distributions, we assume the time delay distributions are right-skewed. We consider lognormal, Weibull, and gamma distributions. Each distribution is parameterised by two variables, $\theta_1$ and $\theta_2$. For all three distributions, $\theta_1$ is the mean of distribution. We take $\theta_2$ to be the standard deviation for the gamma and lognormal distributions, and the scale parameter for the Weibull distribution. For all time delays and parameter distributions, a flat exponential prior with rate 0.0001 was used for $\theta_2$. For the infection to hospitalisation and ascertainment delays, a normal prior with a mean and standard deviation of 15 was used for $\theta_1$. For length of stay, a normal prior with a mean of 7 and standard deviation of 15 was used for $\theta_1$. The lognormal, gamma and Weibull models were compared using LOO cross validation with Pareto smoothed importance sampling.

## The overall and instantaneous case hospitalisation risk

Similar to the criteria developed for the time delay modelling, for a hospitalisation to be included the case required a valid inpatient stay of a day or more, with either a mpox diagnosis code, a positive test within 21 days of hospitalisation or was positive for the monkeypox virus during their stay. We excluded cases with a PCR specimen date before 1st June 2022 as public health policies changed considerably during this time. Individuals with specimen dates within the estimated 95th percentile for the time from infection to hospitalisation distribution were removed and we aimed to exclude cases that may not have yet sought hospital treatment. It was unnecessary to exclude patients still in hospital for the CHR calculation provided they met the described definition. In total, this yielded 3375 cases and 172 hospitalisations for our CHR analysis.

The overall sample CHR is based on all cases included in our study. To calculate uncertainty, we assume binomial credible intervals based on the total number of cases with a beta distribution prior. The overall sample CHR was further calculated for each age group and sex. For the instantaneous CHR, we consider two variations, weekly and daily. For weekly, we aggregate cases each week and calculate the proportion of cases that go to hospital, with credible intervals generated through binomial uncertainty with a beta distribution prior. For the daily CHR, we consider each patient-level data point as a sample from a Bernoulli random variable, where '1' is hospital admission and '0' is no hospital admission. The probability of this random variable reflects the CHR. To simultaneously capture the daily time varying trend and uncertainty, we fit a logistic generalised additive model with a thin plate spline through time. We implement this using the mgcv package[39], with 23 knots, a logit link function, and Bernoulli error.

## Infection hospitalisation risk

We developed a model to infer the risk of hospitalisation given infection with the monkeypox virus, referred to as the IHR. In previous sections, we discussed the CHR, which estimates the risk that a case will be hospitalised, given that their infection is identified. The central challenge in estimating the IHR arises from the fact that we usually capture only a subset of infections from a virus in testing data and therefore, in the absence of robust serological and prevalence studies we must estimate the ascertainment of infections.

To estimate the IHR we have developed a two-step methodology. Firstly, we inspect the bias in the CHRs of different subgroups, conditional upon whether they were notified of their exposure. An index infection is defined, for this paper, as an infected individual that did not receive a notification of their exposure to the monkeypox virus, and an index case is an individual that did not receive a notification of their exposure and was identified through their presentation to a clinician or healthcare provider. As a consequence, index cases are, on average, more severe. Conversely, a secondary infection is defined as an infected individual who was notified of their exposure, and if a secondary infection was subsequently tested for the monkeypox virus then the individual became a secondary case. Secondary cases, consequently, have a higher ascertainment rate (and a lower CHR). By comparing the CHRs in different subpopulations, we can constrain the range of plausible values for the IHR, in addition to constraining the ascertainment rates of each subpopulation. This allows us to define a prior distribution of plausible values, which we refer to as the ascertainment bias prior.

Once we have obtained the ascertainment bias prior distributions, we develop a full model including the incidence and ascertainment rates for each subpopulation, in combination with the infection-associated hospital admissions. We use our estimated infection to ascertainment/hospitalisation delay distributions to fit to the observed cases/hospitalisations over time. This allows us to further inform the ascertainment rates of each group over time; if we have high ascertainment rates of infections, then we ought to be able to reliably predict hospitalisations. Conversely, low ascertainment of infections would make it difficult to predict future hospitalisation conditional upon the observed number of infections.

For a given infection, let $A = \{\text{ascertained}\}$ be the event that they are ascertained, and let $H = \{\text{hospitalised}\}$ be the event that infection leads to hospitalisation. Our main quantity of interest is the IHR $P(H)$. Let $S \in \{p,s,u\}$ denote which subpopulation (index, secondary, untraced) a given case belongs to. Index cases, $p$, are cases that sought out a test from a clinician or healthcare services to diagnose their infection. Secondary cases are those who were made aware of their exposure to the monkeypox virus via successful contact tracing attempts in our data. Untraced cases are cases who do not appear in our contact tracing data.

For untraced cases, it is unknown whether they were aware of their exposure or not. As a result, whilst we do not have records of these individuals being notified of their exposure, it is possible that they were made aware of their exposure through methods such as partner notification, therefore we should explore the possibility that a substantial portion of untraced cases can effectively be considered index cases. We observe that the CHR of untraced cases is significantly lower than the CHR of index cases and is much closer to the CHR of secondary cases. As a result, it is necessary to consider a higher case ascertainment rate of untraced cases than that of index cases, and we propose partner notification as a reasonable explanation of why this may be the case. We provide an additional model fit in our supplementary materials where we omit the untraced group, however doing so results in a significantly decreased sample size, and as a result higher uncertainty in the posterior estimate of the IHR.

We will parameterise our model in terms of the IHR, $P(H)$, and the probability of ascertaining a non-hospitalised infected member of a subpopulation, $P(A|\neg H,S)$. Additionally, we assume we capture all hospitalised cases, $H \Rightarrow A$, and that the probability of hospitalisation is independent of subpopulation, $H \perp S$. Together, this provides the following expression for the CHR for subpopulations $S \in \{p,s\}$;

$$P(H|A,S) = \frac{P(H,A,S)}{P(A,S)} = \frac{P(H,A|S)P(S)}{P(A|S)P(S)} = \frac{P(A|H,S)P(H)}{P(A|H,S)P(H)+P(A|\neg H,S)P(\neg H)}$$
$$= \frac{P(H)}{P(H)+P(A|\neg H,S)P(\neg H)}. \qquad (10)$$

For the untraced subpopulation, we will assume that it is a mixture of either index or secondary cases, however for a given member we do not know which group. As a result, we will additionally estimate the probability that a member of the untraced population is an index or secondary case.

Let $U \in \{p,s\}$ be the event that a member of the untraced population is either an index or secondary member. Then we have that

$$P(H|A,S = u) = P(H|A,S = p)P(U = p) + P(H|A,S = s)P(U = s), \qquad (11)$$

where $P(U = p)$ is a parameter to be estimated.

Let $N_{H,S} \in \mathbb{Z}_+$ be the total number of hospitalisations for a subpopulation, and $N_{A,S} \in \mathbb{Z}_+$ be the total number of ascertained cases for that subpopulation. Then we can estimate the CHR for that subpopulation using a binomial model

$$N_{H,S} \sim \text{Binomial}(N_A, P(H|A,S)), \qquad (12)$$

where $P(H|A,S)$ is the CHR. We then fit a model that estimates $P(H)$ (the IHR), and $P(A|\neg H,S)$ for each subpopulation, and this forms our prior distributions for the full model. Importantly, the prior reduces the space of possible solutions, however the solutions are not fully identified; for example, the prior is only able to estimate the ratio of ascertainment between groups. We provide visualisations of the obtained prior distribution in our supplementary materials, and comparisons between the prior and the posterior distribution.

We now define our full model, where in addition to estimating the IHR and ascertainment rate, we will also estimate the incidence rate of different subpopulations over time. We model $T \in \mathbb{N}$ days of data, letting $x_{t,s'} \in \mathbb{Z}_+$ and $h_{t,s'} \in \mathbb{Z}_+$ be the number of new ascertained cases / hospitalisations respectively on day $t \in [1,\ldots,T]$ for subpopulation $S = s'$.

As model inputs, we provide case ascertainment and hospitalisation delay distributions. These are defined using the posterior medians we obtain from the time delay models. Let $\delta_t^{(h)} \in [0,1]$ be the probability that a case is hospitalised $t$ days after they were infected, conditional on being a hospitalised case, and let $\delta_t^{(a)} \in [0,1]$ be the probability that a case is ascertained $t$ days after they were infected, conditional on being an ascertained case. Both $\delta^{(a)}, \delta^{(h)}$ are $T$-simplexes, that is, they are vectors of length $T$ that sum to 1.

In addition to the model parameters used in the prior distribution, the full model contains time varying incidence rates for each subpopulation. Let $\theta_t^{(s')}$ be the expected number of new infections on day $t$ for subpopulation $S = s'$. We assume that the incidence rates are independent for all populations.

For brevity, we let $P(H) = p_h$, and $P(A|\neg H,S) = \alpha_S$. Let $\lambda_{t,t'}^{(a,s')}, \lambda_{t,t'}^{(h,s')} \in \mathbb{R}_+$ be the expected number of infections who are infected on day $t$ and ascertained/hospitalised respectively on day $t'>t$, for a given subpopulation $S = s'$. We have that

$$\lambda_{t,t'}^{(h,s')} = \theta_t^{(s')} \cdot \delta_{t'-t}^h \cdot p_h, \qquad (13)$$

and

$$\lambda_{t,t'}^{(a,s')} = \theta_t^{(s')} \cdot \delta_{t'-t}^{(a)} \cdot P(A|S = s') = \theta_t^{(s')} \cdot \delta_{t'-t}^{(a)} \cdot (\alpha_S(1 - p_h) + p_h). \qquad (14)$$

Let

$$\mu_{t'}^{(a,s')}, \mu_{t'}^{(h,s')} \in \mathbb{R}_+ \qquad (15)$$

be the expected number of ascertained/hospitalised cases on day $t$, obtained via $\mu_{t'}^{(a,s')} = \sum_{t=1}^{t'} \lambda_{t,t'}^{(a,s')}$ and $\mu_{t'}^{(h,s')} = \sum_{t=1}^{t'} \lambda_{t,t'}^{(h,s')}$.

Given that we now have values for the expected number of cases/hospitalisations over time, it remains for us to fit to the data. Before this, we adjust for day-of-week reporting effects in case ascertainment

and hospitalisations over time for each subpopulation. Let $\beta_d^{(a,s')} \in \mathbb{R}$ be the day of week effect for case ascertainment in subpopulation $s'$ for $d \in [0,\dots,6]$. We obtain the day-of-week adjusted expected number of cases in subpopulation using

$$\widetilde{\mu_t^{(a,s')}} = \exp\left(\log\left(\mu_t^{(a,s')}\right) + \beta_{t\,\mathrm{mod}7}^{(a,s')}\right). \tag{16}$$

A similar transformation is used to obtain the day of week adjusted number of hospitalisations for each subpopulation.

For the untraced subgroup, the day of week effect appears to change partway through the time series. As a result, we fit a day-of-week effect to the period from 16th May 2022 to 13th July 2022, and a second independent day-of-week effect for the period 14th July 2022 to 22nd September 2022.

Finally, to model the observed time series, we assume the data is generated using a negative binomial distribution; $x_{t,s'} \sim \mathrm{NegBin}\left(\widetilde{\mu_t^{(a,s')}}, \phi_{a,s'}\right)$ and $h_{t,s'} \sim \mathrm{NegBin}\left(\widetilde{\mu_t^{(h,s')}}, \phi_{h,s'}\right)$, where $\phi_{a,s'}, \phi_{h,s'} \in \mathbb{R}_+$ are the overdispersion parameters.

For the ascertainment rate and hospitalisation risk priors, we use the posteriors obtained from the ascertainment bias prior. For the incidence rate, we estimate the incidence rate in 10-day intervals, where the evolution of the incidence rate between each interval is controlled by a second order random walk smoothing prior. For the overdispersion parameters, we follow the standard practice[40] of assuming that $\frac{1}{\phi} \sim N(0,1)$. For all other parameters, we assumed flat uninformative priors.

### Computational details
Analysis of the data was conducted in R version 4.3.2. The models for estimating the time delay distributions and estimating the IHR were implemented using cmdstanr[41] (version 0.6.1), and Bayesian computation was performed using Hamiltonian Markov Chain Monte Carlo. Convergence was assessed using potential scale reduction factor or $\hat{R}$ where a value less than 1.01 is desirable. For the IHR model 4 chains were used to draw 2000 samples from the posterior, with the first 1000 samples being discarded as burn-in for the MCMC sampler.

### Reporting summary
Further information on research design is available in the Nature Portfolio Reporting Summary linked to this article.

## Data availability
The data used in this study is not publicly available. UKHSA operates a robust governance process for applying to access protected data that considers: the benefits and risks of how the data will be used; compliance with policy, regulatory and ethical obligations; data minimisation; how the confidentiality, integrity, and availability will be maintained; retention, archival, and disposal requirements; best practice for protecting data, including the application of 'privacy by design and by default', emerging privacy conserving technologies and contractual controls.Access to protected data is always strictly controlled using legally binding data sharing contracts.UKHSA welcomes data applications from organisations looking to use protected data for public health purposes. To request an application pack or discuss a request for UKHSA data you would like to submit, contact DataAccess@ukhsa.gov.uk.

## Code availability
The Stan code to model the infection hospitalisation risk and the doubly interval censored model adjusted for right truncation is provided in the Supplementary Code file.

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

## Author contributions

TW conceived and led the study. TW, RSP, MF and CEO developed the methods and code for the time delay models. TW, MF, CEO, RSP, and RC developed the data criteria and visualisation code. TW, RSP, CEO, and MF developed the methods and code for the case hospitalisation risk models. TW and MF developed the methods for the infection hospitalisation risk model. Thomas Ward, Christopher E Overton, and Martyn Fyles wrote the original manuscript. TW, FC, RSP, MF and CEO reviewed the manuscript. TW, MF, and CEO wrote the revisions.

## Competing interests

The authors declare no competing interests.

### Ethical approval

UKHSA have an exemption under regulation 3 of section 251 of the National Health Service Act (2006) to allow identifiable patient information to be processed to diagnose, control, prevent, or recognise trends in communicable diseases and other risks to public health.
