## [Peer Review File · Nature Communications]

Understanding the infection severity and epidemiological characteristics of mpox in the UKREVIEWER COMMENTS

Reviewer #1 (Remarks to the Author):

I previously reviewed this manuscript for a different journal.

Although I can not fully check if changes have been made from that submission my review and comparison of the two proofs suggests there have been few, if any, substantial changes and hence many of my concerns remain unaddressed.

Comments:

The authors conduct an analysis of the average length of stay and the hospitalisation risk for mpox in the UK.

Strengths of the paper include a large dataset, access to hospital-episode-statistics data and a strong bayesian approach.

Introduction:

1) Line 14: . Analysis of 122 mpox polymerase chain reaction (PCR) tests found the average age had risen to 29 and 69% of cases were male [5].

This needs a lot of clarification. These are different settings with different clades of virus and plausibly different transmission networks. In addition the point not made is in the different proportions who would have been smallpox vaccinated at the time of these two studies, which likely is a major driver of a change in age.

2) A considerable proportion of transmission within this outbreak has been found to have occurred in the presymptomatic phase [7]. - Reference 7 is a modelling study. Whilst it does suggest that transmission occurred during presymptomatic phase I'm not sure it is correct to say that this proves it as currently written.

3) A fundamental issue with the paper is that the rationale and scope of admissions has changed considerably. Initially all uk cases were hospitalised regardless of clinical need, then a model was adopted whereby some cases were hospitalised for clinical need and some for isolation purposes (with the rest being managed as outpatients). This was a UK specific approach and different approaches were used in other settings. As such the number of admissions in the UK has been heavily dependent on the approach taken to isolation as much as to the clinical severity of the cases. I can not see this discussed anywhere in the paper and this makes interpreting data on hospitalisation rates difficult. What proportion of admissions fell into each category? How does the average length of stay vary by type? Does the proportion of admissions in each category change over time? Does their length of stay?

For example in the discussion you state:

he 2022 outbreak of mpox has seen considerable inter-country heterogeneity between estimates of the CHR, which is a consequence of varying study periods and the ascertainment of cases. Whilst I am sure these are drivers the different CHRs almost certainly reflect different policies with regards to admission in different settings. The nationally reported admission rate in Spain is much lower than in the UK reflecting that they admitted far fewer patients for isolation for example. The discussion does not really adequately address the extent to which public health policies for a disease like mpox shape the admission rate.

4) It appears that you exclude individuals still in hospital after a specific date - may this have resulted in individuals with particularly prolonged admissions being excluded? There were relatively few new cases of mpox in the UK by the 26th October so I am surprised that of 600+ admissions only c.150 contribute to your length of stay data.

5) Whilst the model is interesting it lacks a lot of clinical information to really help make public health decisions. How does risk vary by age? By HIV status? By CD4 count. These are all likely key determinants of severity and hospitalisation risk but are not accounted for nor is there discussion of this in the limitations.

Reviewer #2 (Remarks to the Author):

This paper reports analysis results of some outcomes of Mpox in UK. The Bayesian analysis method used for the data analysis is completely parametric, so that the data analysis results will be dependent on the selected prior and likelihood distributions. This should be stated as a weakness of the study. Following are specific comments.

LL71-72: "... patients were excluded if they were still in hospital on the 72 26th October.": Those patients should have been included by using censored data analysis for length of stay.

LL73-75: "To calculate the time from infection to hospitalisation, patients that reported symptom onset after hospitalisation were excluded and for those that had multiple admissions the earliest admission date 75 was used.": Those patients should have been included by using left truncated data analysis.

Comments to Manuscript: Understanding the Severity and Clinical Characteristics of Mpox

This manuscript considered estimation of the overall and instantaneous case hospitalization risk (CHR) and the infection hospitalization risk (IHR), through Bayesian modeling. Specifically, a Bayesian doubly interval censored model adjusted for right truncation was used to calculate mpox length of hospital stay and the time from infection to hospital admission. Furthermore, a Bayesian model was used adjusting for the ascertainment of cases through contact tracing.

The paper is really hard to read, since many of the mathematical terms and formulas are not well defined when they were first used. Therefore, the reader would need to read between lines to “guess” what are the meanings of the variables and how the formulas were derived. It has greatly hurt the readability and scientific merit of the manuscript. Here I’m only listing a few of those issues. For example, key variables H , T , and D are never defined when they were first used on Line 112; and I only found the definition of H in the next page (line 139). In addition, it seems like the symptom onset observation interval $[s_1, s_2]$ should be on the calendar time scale, but $S \in [s_1, s_2]$ is defined as the maximum incubation period, which should be the time difference between the infection time and the symptom onset time. Moreover, the same term S was used as both the maximum incubation period and index of subpopulation in this manuscript.

Here are my other comments

- In the introduction, it is not clear to me what is the difference between the “case hospitalization risk (CHR)” and “infection hospitalization risk (IHR)”. Indeed, it is only clear when I read till Line 201-203.
- It’s not totally clear to me how many of the 694 patients with a hospital episode were excluded from the length of stay calculation due to (1) they did not have an inpatient stay, (2) they were still in hospital on the 26th October, or (3) missing data? It is hard

to argue that the remaining 168 patients are representative of the 694 patients with a hospital episode.

- Line 112: H , T , and D are never defined.
- Line 112. Is it not clear to me how this equality is obtained. S is defined as the maximum incubation period (line 108). I don't understand why it should belong to the for symptom onset time interval $[s_1, s_2]$.
- Other minor comments
 - Line 31: “within defined a temporal window”. Should be “within a defined temporal window”?

REVIEWER COMMENTS

Reviewer #1 (Remarks to the Author):

I previously reviewed this manuscript for a different journal. Although I cannot fully check if changes have been made from that submission my review and comparison of the two proofs suggests there have been few, if any, substantial changes and hence many of my concerns remain unaddressed.

We thank the reviewer for their time and comments below.

Comments:

The authors conduct an analysis of the average length of stay and the hospitalisation risk for mpox in the UK. Strengths of the paper include a large dataset, access to hospital-episode-statistics data and a strong bayesian approach.

Introduction:

1) Line 14: . Analysis of 122 mpox polymerase chain reaction (PCR) tests found the average age had risen to 29 and 69% of cases were male [5]. This needs a lot of clarification. These are different settings with different clades of virus and plausibly different transmission networks. In addition the point not made is in the different proportions who would have been smallpox vaccinated at the time of these two studies, which likely is a major driver of a change in age.

Thank for the comment. The primary distinction was transmission routes, but we have now amended the language here to clarify this point and discussed the vaccine uptake bias. "The age composition of mpox cases has been influenced by the circumstances of the outbreaks, changes in transmission routes, and vaccination campaigns."

2) A considerable proportion of transmission within this outbreak has been found to have occurred in the presymptomatic phase [7]. - Reference 7 is a modelling study. Whilst it does suggest that transmission occurred during presymptomatic phase I'm not sure it is correct to say that this proves it as currently written.

We thank the reviewer for this comment, and we have amended the language here. Since this study was published other studies have found similar findings, and we have included some of these references in the paper. The referenced study also provides individual level case records linked through personally identifiable information as well as the modelling component.

3) A fundamental issue with the paper is that the rationale and scope of admissions has changed considerably. Initially all uk cases were hospitalised regardless of clinical need, then a model was adopted whereby some cases were hospitalised for clinical need and some for isolation purposes (with the rest being managed as outpatients).

The hospitalisation of cases for isolation was only a policy adopted for the first detected cases and they have not been included in our sample used for analysis. Further context has been included on this "In the UK, the early CHR was biased by a policy of clinical isolation of positive cases, independent of clinical need. However, this affected the early stages of the outbreak, and this policy was removed after the outbreak became established. Therefore, in this paper we consider a study period of 01 June 2022 to 30 September 2022."

This was a UK specific approach and different approaches were used in other settings. As such the number of admissions in the UK has been heavily dependent on the approach taken to isolation as much as to the clinical severity of the cases. I can not see this discussed anywhere in the paper and this makes interpreting data on hospitalisation rates difficult. What proportion of admissions fell into each category? How does the average length of stay vary by type? Does the proportion of admissions in each category change over time? Does their length of stay?

The hospitalisation of cases for isolation was only a policy adopted for the first detected cases and they have not been included in our sample used for analysis. We have included further context in the paper (see above).

For example in the discussion you state:

The 2022 outbreak of mpox has seen considerable inter-country heterogeneity between estimates of the CHR, which is a consequence of varying study periods and the ascertainment of cases. Whilst I am sure these are drivers the different CHRs almost certainly reflect different policies with regards to admission in different settings. The nationally reported admission rate in Spain is much lower than in the UK reflecting that they admitted far fewer patients for isolation for example. The discussion does not really adequately address the extent to which public health policies for a disease like mpox shape the admission rate.

Further context has been provided in the discussion on the policy implications for admissions and how this pertains to the data used in this study. "There has been considerable inter-country variation in estimates of the CHR across the 2022 outbreak of mpox, which will have been influenced by differences in public health policy. These distinctions in policy will have an impact on the case ascertainment rate and the criteria for hospital admission."

4) It appears that you exclude individuals still in hospital after a specific date - may this have resulted in individuals with particularly prolonged admissions being excluded? There were relatively few new cases of mpox in the UK by the 26th October so I am surprised that of 600+ admissions only c.150 contribute to your length of stay data.

Most of the individuals that were removed had a length of stay of zero, having been assessed at hospital for diagnostic purposes and not admitted for an inpatient stay. We have now updated the data processing section to provide a detailed description of the number of individuals excluded by each criterion. Missing discharge dates were excluded since SUS data only includes complete hospital stays, so we know these individuals have been discharged and the discharge date is missing (rather than the hospital stay being right-censored).

5) Whilst the model is interesting it lacks a lot of clinical information to really help make public health decisions. How does risk vary by age? By HIV status? By CD4 count. These are all likely key determinants of severity and hospitalisation risk but are not accounted for nor is there discussion of this in the limitations.

We thank the reviewer for this comment. The reviewer raises an interesting point and we have therefore, included the CHR by age groups in the supplementary information and plotted the age distribution of admissions, with no significant variation observed across age groups. We have included further context in the limitations of the study. We did not have an adequate sample size in the admission data to provide a meaningful modelled estimate for age stratifications of the IHR (HIV status and CD4 was not available to the UKHSA team for all admissions data).

Reviewer #2 (Remarks to the Author):

This paper reports analysis results of some outcomes of Mpox in UK. The Bayesian analysis method used for the data analysis is completely parametric, so that the data analysis results will be dependent on the selected prior and likelihood distributions. This should be stated as a weakness of the study. Following are specific comments.

We are thankful to the reviewer for their time. We have added a paragraph to the limitations mentioning the parametric nature of the time delay distributions used.

LL71-72: "... patients were excluded if they were still in hospital on the 26th October.": Those patients should have been included by using censored data analysis for length of stay.

Missing discharge dates were excluded since the SUS data only includes complete hospital stays, so we know these individuals have been discharged and the discharge date is missing (rather than the hospital stay being right-censored). Therefore, these patients are missing dates rather than genuinely still being in hospital. We agree that where possible, right censoring corrected analysis is stronger than right truncation corrected analysis, however, this relies on the censored data being reliable, but here it appeared to be a data quality issue. We have clarified this in the paper to make it clear these were excluded for data quality reasons rather than right-censoring.

LL73-75: "To calculate the time from infection to hospitalisation, patients that reported symptom onset after hospitalisation were excluded and for those that had multiple admissions the earliest admission date 75 was used.": Those patients should have been included by using left truncated data analysis.

This was a consequence of incorrect symptom questionnaire data. Therefore, this data was excluded. We have clarified this in the paper.

Reviewer #3 - please see attached document

We would like to thank the reviewer for their time reviewing this manuscript.

Comments to Manuscript: Understanding the Severity and Clinical Characteristics of Mpox

This manuscript considered estimation of the overall and instantaneous case hospitalization risk (CHR) and the infection hospitalization risk (IHR), through Bayesian modeling. Specifically, a Bayesian doubly interval censored model adjusted for right truncation was used to calculate mpox length of hospital stay and the time from infection to hospital admission. Furthermore, a Bayesian model was used adjusting for the ascertainment of cases through contact tracing.

The paper is really hard to read, since many of the mathematical terms and formulas are not well defined when they were first used. Therefore, the reader would need to read between lines to guess what are the meanings of the variables and how the formulas were derived. It has greatly hurt the readability and scientific merit of the manuscript. Here I'm only listing a few of those issues. For example, key variables H, T, and D are never defined when they were first used on Line 112; and I only found the definition of H in the next page (line 139). In addition, it seems like the symptom onset observation interval $[s_1; s_2]$ should

be on the calendar time scale, but $S_2 [s_1; s_2]$ is defined as the maximum incubation period, which should be the time difference between the infection time and the symptom onset time. Moreover, the same term S was used as both the maximum incubation period and index of subpopulation in this manuscript.

The methods section has now been amended to make the definitions of the notation clearer.

Here are my other comments

_ In the introduction, it is not clear to me what is the difference between the case hospitalization risk (CHR)" and infection hospitalization risk (IHR)". Indeed, it is only clear when I read till Line 201-203.

In the introduction the CHR is now defined on line 23-24 the IHR is defined line 40-41.

_ It's not totally clear to me how many of the 694 patients with a hospital episode were excluded from the length of stay calculation due to (1) they did not have an inpatient stay, (2) they were still in hospital on the 26th October, or (3) missing data? It is hard 1 to argue that the remaining 168 patients are representative of the 694 patients with a hospital episode.

We excluded individuals that were not admitted to hospital for an inpatient stay. Most of the individuals were assessed for diagnostics purposes but not admitted for treatment. We now provide a breakdown of the numbers excluded at each stage.

_ Other minor comments

{ Line 31: "\within defined a temporal window". Should be "\within a defined temporal window"?

Thank you, this has now been amended.

REVIEWERS' COMMENTS

Reviewer #1 (Remarks to the Author):

I note the changes.

The authors state that the isolation policy was lifted by June. This is not really correct. Initially ALL cases were isolated. After that the UK adopted a classification process where people were grouped into

- Admit for clinical need (I think this was called Group A if I recall correctly)
- Admit as cant isolate at home (Group B)
- Isolate at home

This inevitably still resulted in a higher isolation rate compared to other countries because Group B was subjective (compare UK admission rate with Spain for example).

As such I still dont really feel this limitation is adequately addressed. The paper really needs to present the data on what proportion of patients were admitted across Cat A and B to make sense of this data.

Reviewer #2 (Remarks to the Author):

(none)

Reviewer #3 (Remarks to the Author):

what do you mean by "to avoid right-truncation" on line 70? The resulting data still have right-truncation.

From line 75-82 and Table 1, it still reads a bit confusing that 155 patients were included in the analysis out of a total 824 individuals. I would suggest that the authors report the 172 cases with mpox determination and inpatient stay of at least a day (this is the population of interest) instead of (or in addition to) the 824 individuals with hospitalization code in this paragraph. Base the analysis on 155 cases out of 172 is more convincing as a representative analysis.

Line 80: "patients were excluded if the discharge date was missing". Can you please add your explanation on missing discharge date in the rebuttal letter to the manuscript?

Remaining reviewer comments

	REVIEWERS' COMMENTS
Remaining reviewer comments: Reviewer #1: I note the changes. The authors state that the isolation policy was lifted by June. This is not really correct. Initially ALL cases were isolated. After that the UK adopted a classification process where people were grouped into  - Admit for clinical need (I think this was called Group A if I recall correctly) - Admit as cant isolate at home (Group B) - Isolate at home This inevitably still resulted in a higher isolation rate compared to other countries because Group B was subjective (compare UK admission rate with Spain for example). As such I still dont really feel this limitation is adequately addressed. The paper really needs to present the data on what proportion of patients were admitted across Cat A and B to make sense of this data. Reviewer #2: - Reviewer #3: what do you mean by "to avoid right-truncation" on line 70? The resulting data still have right-truncation. From line 75-82 and Table 1, it still reads a bit confusing that 155 patients were included in the analysis out of a total 824 individuals. I would suggest that the authors report the 172 cases with mpox determination and inpatient stay of at least a day (this is the population of interest) instead of (or in addition to) the 824 individuals with hospitalization code in this paragraph. Base the analysis on 155 cases out of 172 is more convincing as a representative analysis. Line 80: "patients were excluded if the discharge date was missing". Can you please add your explanation on missing discharge date in the rebuttal letter to the manuscript?	Reviewer #1 (Remarks to the Author): I note the changes. We thank the reviewer for their further review of the paper. The authors state that the isolation policy was lifted by June. This is not really correct. Initially ALL cases were isolated. After that the UK adopted a classification process where people were grouped into  - Admit for clinical need (I think this was called Group A if I recall correctly) - Admit as cant isolate at home (Group B) - Isolate at home This inevitably still resulted in a higher isolation rate compared to other countries because Group B was subjective (compare UK admission rate with Spain for example). As such I still dont really feel this limitation is adequately addressed. The paper really needs to present the data on what proportion of patients were admitted across Cat A and B to make sense of this data. We have been informed by the NHS that the policy of admission to hospital due to an individual being unable to isolate at home was an extremely small proportion of cases that presented themselves to hospital. And that the HCID official derogation date (of the policies described above) was the 5/7/22 (please see https://www.england.nhs.uk/long-read/update-on-high-consequence-infectious-disease-hcid-status-of-mpox/). We have been informed that data on the admission categories defined above was not recorded in the digital patient records, so we cannot provide the exact proportions. We have therefore included further context in the limitations of the study to convey this. In our analysis, we restricted the dates to where the instantaneous CHR became roughly constant, so although this has some overlap with the HCID period, the impact of these cases should be very limited. Reviewer #2 (Remarks to the Author): (none) Reviewer #3 (Remarks to the Author): We thank the reviewer for their time.

what do you mean by “to avoid right-truncation” on line 70? The resulting data still have right-truncation.

This has been removed.

From line 75-82 and Table 1, it still reads a bit confusing that 155 patients were included in the analysis out of a total 824 individuals. I would suggest that the authors report the 172 cases with mpox determination and inpatient stay of at least a day (this is the population of interest) instead of (or in addition to) the 824 individuals with hospitalization code in this paragraph. Base the analysis on 155 cases out of 172 is more convincing as a representative analysis.

Line 80: “patients were excluded if the discharge date was missing”. Can you please add your explanation on missing discharge date in the rebuttal letter to the manuscript?

We have amended the description in the methods section to make this clearer for the reader.